# PRRSV-1 Stabilization Programs in French Farrow-to-Finish Farms: A Way to Reduce Antibiotic Usage

**DOI:** 10.3390/ani13142270

**Published:** 2023-07-11

**Authors:** Charlotte Teixeira Costa, Pauline Berton, Gwenaël Boulbria, Valérie Normand, Mathieu Brissonnier, Arnaud Lebret

**Affiliations:** 1Rezoolution, ZA de Gohélève, 56920 Noyal-Pontivy, Francea.lebret@rezoolution.fr (A.L.); 2Porc.Spective, ZA de Gohélève, 56920 Noyal-Pontivy, France

**Keywords:** porcine reproductive and respiratory syndrome virus, health management, antibiotic usage, pig production sector, PRRSV-1 stabilization program

## Abstract

**Simple Summary:**

This study reports an investigation into 19 French farrow-to-finish farms that successfully implemented the porcine reproductive and respiratory syndrome virus type 1 (PRRSV-1) stabilization protocol between 2007 and 2019. One year before (P1) and one year after (P2) the introduction of this protocol, the antibiotic consumptions (expressed in mg/PCU and ALEA) were compared for each farm. The difference between P1 and P2 was also calculated in percentages. The transition from P1 to P2 revealed that the higher the consumption levels were in P1, the greater the antibiotic reduction in P2. These results emphasize how stability against PRRSV-1 seems to reduce antibiotic usage, particularly in farms with a high level of consumption.

**Abstract:**

Infection with the porcine reproductive and respiratory syndrome virus type 1 (PRRSV-1) has serious economic consequences for the pig industry. Swine practitioners and other agricultural advisors often describe an increase in antibiotic use when PRRSV-1 is circulating. Our objective was to assess the impact of PRRSV-1 stabilization programs on reducing antibiotic use in 19 French farrow-to-finish farms that successfully implemented such a protocol between 2007 and 2019. For each farm, we compared the global antibiotic consumption, including all physiological stages (expressed in mg/PCU and ALEA) one year before (P1) and one year after (P2) the implementation of the protocol, and the change between P1 and P2 was calculated in percentages. The data were also analyzed by level of consumption. We showed that antibiotic use decreased significantly between P1 and P2 if expressed in mg/PCU and showed a decreased tendency in terms of exposure (ALEA) after PRRSV-1 stabilization. Concerning the change from P1 to P2, depending on the level of consumption in P1, our results showed that the higher the consumption levels were in P1, the greater the antibiotic reduction in P2. This study highlights the ability of a stabilization protocol against PRRSV-1 to reduce antibiotic use, especially on farms that have high consumption rates. These hopeful results show that further investigations about the relationship between PRRSV-1 and antibiotic usage could be beneficial.

## 1. Introduction

Porcine reproductive and respiratory syndrome virus (PRRSV) is classified within the genus *Arterivirus* and causes enzootic viral disease. In the west of France—and particularly in Brittany, where nearly 70% of French pork production is concentrated—it affects more than 60% of farms, with many respiratory and reproductive problems, and generates serious economic consequences [1,2,3]. Within the *Arterivirus* family, the species *Betaarterivirus suid* 1 (PRRSV-1) are most widespread in Europe, while the *Betaarterivirus suid* 2 are mainly present in North America and Asia [4,5]. Regarding the impact of PRRSV-1 circulation in herds on antibiotic consumption, agricultural advisors have noted an increase in antibiotic usage when PRRSV-1 is present, but few reports support this observation. However, the observation makes sense in view of bacterial co-infections that frequently require antibiotic treatments [6,7].

For several years, PRRSV stabilization protocols have been evaluated and implemented, mainly in North America, using mass vaccination and improvements in biosecurity measures [8,9]. In France, since the 2000s, combining the mass vaccination of sows and their piglets with a modified-live virus (MLV) vaccine, together with closure of the farm and unidirectional pig and human flows, has been widely implemented [10].

The success of such protocols is established by the demonstration of the absence of viral circulation in the breeding herd. Based mainly on blood samples in weaning piglets, sampling protocols are described in the American Association of Swine Veterinarians (AASV) guidelines [11]. This permits the classification of farms as: naïve, stable, or unstable (with different prevalence levels).

Measuring antibiotic use is not an easy task. First, antibiotics can be used in different ways; treatments can be administered individually or, when many animals need to be treated, group treatments are often provided orally via feed or water [12]. All characteristics related to antibiotic usage should be considered, e.g., treatment type (metaphylactic or prophylactic), antibiotic class, route of administration (injectable, premix, oral solutions). Secondly, there are various methods for quantifying usage. Many indicators are available to measure antimicrobial use in veterinary medicine and the results can differ substantially depending on the method used [13]. Therefore, the selection of the appropriate indicators of antibiotic usage requires a clear objective that will permit reliable comparisons through standardized parameters.

In the global effort to combat antimicrobial resistance, reducing antibiotic use is a key point of improvement on swine farms. Several studies have already shown some useful measures for reducing antibiotic usage, e.g., biosecurity measures [14]. In France, two initiatives—named Plan ‘Ecoantibio’ 1 and 2—have been launched to address the recommendations on antibiotic use by the World Organization for Animal Health (WOAH), the World Health Organization (WHO), and the United Nations Food and Agriculture Organization (FAO). The first plan (EcoAntibio-1) induced a decrease of 41.5% in antibiotic consumption in pigs between 2011 and 2016 [15]. With the second plan (EcoAntibio-2), which aimed to keep up efforts and analyze any issues in the first plan, antibiotic use decreased even further. The overall level of antibiotic exposure in the pig production sector has decreased by 55.5% since 2011.

To the best of our knowledge, no study on the implementation of PRRSV-1 control measures, particularly by way of stabilization programs, has shown an impact on antibiotic use. The aim of our study was to evaluate the impact of PRRSV-1 stabilization protocols on antibiotic usage and to promote these programs with tangible information for practitioners.

## 2. Materials and Methods

### 2.1. Study Design

#### 2.1.1. Selected Farms

With the objective of having sufficient farms to perform statistical analysis, our study included 20 farrow-to-finish farms. They were selected on the basis of the following criteria:(i)Farms should have implemented a successful PRRSV-1 stabilization program. This consisted of mass vaccination of sows and their piglets with a MLV vaccine, together with herd closure, unidirectional pig and human flows, and strict internal biosecurity measures. Regarding the vaccination strategy, it was a combination of a mass vaccination of sows, gilts, and piglets with a PRRSV-1-modified live vaccine (2 times at 1 month interval) followed by batch to batch vaccination of weaning piglets (2 times at 3 weeks interval) lasting around 6 months. All steps of the implementation of PRRSV-1 stabilization programs are described in Berton et al., 2017 [10]. Based mainly on blood samples from weaning piglets, the success of such protocols was confirmed by the absence of viral circulation in the breeding herds following the AASV guidelines [11].(ii)All descriptive data of antibiotics by all administration routes had to be available.(iii)Biomass data should be accessible (for slaughtered pigs and sows).(iv)The prescribing veterinarians and pig farmers had to be the same during both periods (P1 and P2).

For one farm, data were not available; therefore, only nineteen farms were included in the study. The data recorded for each farm included: number of animals treated and potentially treated, years of PRRSV-1 stabilization protocol implementation, batch management, health status, etc.

#### 2.1.2. Antibiotic Data Recorded

All data were collected concerning antibiotic usage one year before the implementation of control measures (P1) and one year following the end of PRRSV-1 monitoring, indicating the success of the plan (P2). The data were recorded from our drug prescription software. These records included the type of drug (antibiotic families and substance administered), number of antibiotic treatments, and routes of administration. The posology (expressed in mg of active ingredient/kg pig weight/day) of each treatment was also recorded to calculate the quantity of antimicrobial active ingredients contained in each treatment. We also recorded whether the administered treatments concerned individual or collective treatment.

#### 2.1.3. Design of the Antibiotic Consumption Categories

The high variability of antibiotic usage in P1 led us to assess the impact of the consumption level in P1 on antibiotic usage reduction after the implementation of a stabilization protocol. To this end, three categories of consumption levels were used for ALEA and two levels for mg/PCU (Table 1).

**Table 1 animals-13-02270-t001:** Categories used to compare antibiotic reduction consumption levels in P1.

Indicators	Categories	Number of Farms	Limits ^1^
ALEA			
	Low	4	<0.5
	Medium	8	0.5–0.9
	High	7	>0.9
mg/PCU			
	Low	10	<100
	High	9	≥100

^1^ Limits used: from David et al., 2021 for ALEA and O’Neill et al., 2020 for mg/PCU [16,17].

### 2.2. Calculation of Indicators

Two indicators were chosen to describe animal exposure to antibiotics and were calculated for each farm. The first one is the ALEA, used by the French authorities to monitor antibiotic sales each year (Formula (1)). It is calculated as follows:(1)$$\mathrm{ALEA}=\frac{\frac{quantity\;of\;active\;ingredient\;\left(mg\right)}{dose\;\left(\frac{\frac{mg}{kg}}{d}\right)\;\times \;duration\;\left(d\right)}}{biomass\;potentially\;treated\;(kg)}$$

Formula (1) considers the number of active substances used (in milligrams), the number of animals treated, the dose, the treatment days, and the current animal weight during therapy (in kilograms). The ALEA value is expressed without a unit. The biomass at risk of being treated is calculated as the number of sows present at the farm and the number of slaughtered pigs produced during both periods multiplied by their theoretical weights. These were calculated using the National Research Institute for Agriculture, Food, and Environment (INRAE) recommendations, namely 105 kg for pigs and 300 kg for sows.

The other indicator chosen is the mg/PCU (Formula (2)), the European indicator used to evaluate antibiotic usage. It takes into account a unique standardized biomass. The main advantage of using biomass is that it allows different animal species to be combined within the same population; this is the approach used by the European Surveillance of Veterinary Antimicrobial Consumption (ESVAC) project to compute the Population Correction Unit (PCU) [18]. It is calculated as follows:(2)$$\mathrm{mg}/\mathrm{PCU}=\frac{quantity\;of\;active\;ingredient\;(mg)}{PCU}$$

Formula (2) considers only the number of active substances used (in mg) and the potential animal weight treated. Calculations of biomass were performed using 65 kg for slaughtered pigs and 250 kg for sows.

### 2.3. Statistical Analysis

All data were recorded using Excel v.22.10 (Microsoft Corporation, Redmond, WA, USA) and calculations were performed with the same software. First, mean comparisons were applied to compare antibiotic usages between P1 and P2 using non-parametric tests (Kruskal-Wallis test) because our data were not normally distributed. Secondly, the change from P1 to P2 was calculated as follows for both indicators (Formula (3)):(3)$$\frac{indicator\;value\;in\;P2-indicator\;value\;in\;P1}{indicator\;value\;in\;P1}$$

Formula (3) gives a percentage of change, calculated for each farm. This percentage was compared between each consumer category determined by the level of consumption in P1 (Table 1). For ALEA, a Wilcoxon test was used to compare the three groups (low, medium, and high levels of consumption). For mg/PCU, only two groups were compared using a Kruskal-Wallis test. For each analysis, the statistically significant difference was set at *p* ≤ 0.05, with 0.05 < *p* ≤ 0.10 considered as a tendency. All statistical analysis was performed using R Studio (v 4.2.2, R Core Team, 2022).

## 3. Results

### 3.1. Features of the Analysed Data

#### 3.1.1. General Characteristics

In total, 19 farrow-to-finish farms located in Brittany (France) were included in the study (Table 2). All of these farms had implemented a successful PRRSV-1 stabilization protocol between 2007 and 2019. The majority of farms (14 out of 19) were managed in a seven-batch management system with weaning at 28 days of age. The number of slaughtered pigs and sows were recorded to calculate the weight of animals potentially treated. The number of sows ranged between 100 and 600 (mean: 273), and the number of fattening pigs ranged between 1762 and 16,662 (mean: 5806).

On all of the farms, piglets were vaccinated against *Mycoplasma hyopneumoniae* and porcine circovirus type 2 (PCV-2). All of the farms purchased their gilts (PRRSV naive) from an external multiplier.

#### 3.1.2. Global Antibiotic Consumption Data

The records of antibiotic consumption allowed the calculation of two indicators: the Animal Level Exposure of Antimicrobials (ALEA), which is the French exposure indicator, and consumption, measured in mg/PCU, which is the European indicator.

The antibiotic classes included fluoroquinolones, aminocyclitols, beta-lactams, trimethoprim and sulphonamides, macrolides, tiamulin and lincomycin, florfenicol, colistin, quinolones, and tetracyclines. Their distribution is described in Figure 1. The antimicrobial classes with the highest weighted treatment occurrence (as a percentage of total) for both periods were tetracyclines, macrolides, tiamulin and lincomycin (referred to as MTL), and colistin.

The antibiotic treatment data were divided between two periods: one period (P1) one year before the implementation of PRRSV-1 control measures, and a second period of the year following the end of PRRSV-1 monitoring, as an indicator of the success of the stabilization protocol (P2). The data concerned a constant number of sows in both periods and, respectively, 110,319 and 116,364 slaughtered pigs in P1 and P2. Considering all of the treatments, a total of 624 g of antibiotics were administered in P1 compared to 607 g in P2, corresponding to a total volume of 1231 g antibiotics. In terms of the absolute quantity used, collective treatments represented almost 30 times that of individual treatments (Table 3).

### 3.2. The Impact of PRRSV-1 Stabilization Programmes

#### 3.2.1. On Overall Antibiotic Consumption

The total consumptions per farm were obtained by summing all of the ALEA per treatment. The same was conducted for mg/PCU. Eleven farms out of 19 had reduced their consumption between P1 and P2 when measured as mg/PCU, and eight when measured as ALEA (Figure 2). On average, we observed a reduction of 0.80 for the ALEA value and 54 mg for the mg/PCU indicator.

Table 2 and P2 (Figure 3). Our analysis showed that the antibiotic usages decreased significantly for mg/PCU, with an average reduction of 4.7 mg/PCU. For ALEA, a downward trend was observed, with a reduction of 0.2.

#### 3.2.2. According to the Level of Antibiotic Consumption in P1

The levels of consumption in P1 seemed to impact the change in usage, both for ALEA and for mg/PCU; the higher the level of consumption in P1, the greater the reduction in antibiotic use in P2 (Figure 4). Indeed, in high consuming herds, the overall level of antibiotic use decreased from 1389.47 mg/PCU to 852.20 mg/PCU, and for the ALEA indicator, from 11.56 to 5.20. For this level of consumption, only one farm increased its use between P1 and P2 for the ALEA indicator, due to a large increase in colistin use, and two farms increased their mg/PCU: one being the farm with increased ALEA values and the other with a huge increase in macrolides, tiamulin, and lincomycin (MTL) use in P2.

When considering the low levels of consumption in P1, as illustrated in Figure 4, it is more difficult to show a substantial reduction in antibiotic usage. Some farms (five out of ten for the mg/PCU value and one out of four for the ALEA value) decreased their usage, while for the others, consumption stayed stable.

Regarding the change in antibiotic consumption (Formula (3), see Section 2.3), the main report in our study concerned high consumers. On average, the farms with a high level of consumption in P1 reduced their usage by 48.8% for mg/PCU and by 13.5% for ALEA. A statistically significant decrease among high consumers was observed compared to low consumers (*p* = 0.006). To a lesser extent, between medium and high consumers, the antibiotic use tended to decrease (*p* = 0.08). However, no significant difference was found between the low and medium consumers (*p* = 0.25). For mg/PCU, farms with a high level of consumption in P1 tended to have a greater reduction in consumption than farms with a low level of consumption in P1 (*p* = 0.07).

## 4. Discussion

In this study, we recorded longitudinal data on antibiotic usages from different farms before and after the implementation of a successful PRRSV-1 stabilization protocol. The PRRSV-1 stabilization programs on our selected farms were implemented between 2007 and 2019. We analyzed these usages using two indicators—the ALEA and mg/PCU values—and also considered the level of consumption in P1. The results of this analysis emphasize the impact of a PRRSV-1 stabilization protocol on antibiotic consumption. These promising results show the importance of further investigations into the relationship between PRRSV-1 and antibiotic usage.

### 4.1. Evaluation of Methods

The ALEA indicator for antibiotic usage is widely used in French analysis. The ALEA values observed in our study are consistent with the averages found by other authors [15,19,20]. This value permits comparisons between countries and periods of time. Different indicators have been proposed and published [13,21,22]. However, unlike most indicators, ALEA is an indicator based on animal course doses administered [18,23]. Differences between nationally established animal-defined daily dosages may have a substantial influence on the results of antimicrobial consumption in swine herds. Thus, harmonized units of measurement and animal weights were used to enable international comparison [24,25,26]. Calculations based on the ALEA indicator are more precise because tonnages sold do not accurately reflect their use. To evaluate the exposure of animals to antibiotics, it is therefore necessary to take into account the dosage, duration of administration, and of course, any change in the number of animals on the farm [27].

Knowledge of the variations in several indicators and the expression of the data using different units provides more precise values of antibiotic consumption. Using various indicators in a complementary approach permits more detailed conclusions to be drawn. In our study, the mg/PCU indicator was used in addition to the ALEA values. This indicator is widely used on a European scale. As described by the UK government: “The mg/PCU is a unit of measurement developed by the European Medicines Agency to monitor antibiotic use and sales across Europe” [28]. It accurately represents the volume of antibiotic usage for standardized animal weights (separate benchmarking values calculated according to different physiological stages).

A limited number of studies have compared several indicators applied to the same antimicrobial usage data in order to achieve the same objective. Some of them investigated the impact of denominator selection when comparing antimicrobial usage based on sales data between countries. For example, Bondt et al. showed that antimicrobial usage, expressed in milligrams of active substance per PCU, overestimated the true difference in usage in The Netherlands compared to Denmark, even though the two countries have similar animal demographics [29]. In our study, we chose this indicator in addition to ALEA in order to guarantee an international scope for our results.

### 4.2. Data Quality

All of the data recorded were based on the data available from our software concerning the delivery of drugs and drugs prescribed by only one veterinarian over the entire analysis period. Indeed, some studies explored the relationship between veterinarian’s prescriptions and antibiotic usage by farmers. For example, a study conducted by Coyne et al. helped to explain the link between the antimicrobial prescribing behaviors of veterinarians and antibiotic usage levels in farms [30]. Another study, by Singer et al., concluded that veterinarians must continue to develop antibiotic control programs to ensure that antibiotics are used only when necessary [31]. Thus, the impact of a veterinarian’s perceptions on antibiotic usage can influence the level of consumption in farms [32,33]. To limit the bias potentially generated by prescribing veterinarians, in our study, only farms followed by the same farm veterinarians during both periods were selected.

As our study was retrospective, some information was lacking to explain the absence of a reduction in some herds. For example, details about the treatment type (e.g., oral vs. individual, antibiotic classes) and usage indications could have allowed us to explain some of the results more accurately. The physiological stages of treated animals were not recorded in our dataset. For this reason, it was not possible to explain some of the increases in antibiotic use in P2. Further studies taking these parameters into account would be helpful.

### 4.3. Antibiotic Usage Changes

In order to reduce antibiotic consumption in breeding herds and to minimize the risks of antibiotic resistance in humans, the European Union implemented a plan to control risks in 2011. EcoAntibio-1 aimed for a reduction of 25% in the total antibiotic consumption between 2012 and 2017. In our study, ten farms carried out their PRRSV-1 stabilization protocols before 2012 and six did so between 2012 and 2017. A second plan (Ecoantibio-2) was implemented between 2017 to 2021, aimed at continuing the goals of the first project. Three of the farms selected for our study had implemented their PRRSV-1 stabilization program during this plan. Our results may have been impacted by the measures of the EcoAntibio plans, but to a minor extent. Indeed, the bias induced by the implementation of these plans should be small as such plans take many years, whereas our comparisons were realized in a limited period. To eliminate this bias, it would be interesting to have a large sample to compare only farms that implemented the stabilization protocol during the same period. However, such a study is not feasible at our practice level.

### 4.4. Classes of Consumption’s Levels

In France, the National Agency for Food, Environment, and Occupational Health and Safety (ANSES) found a 41.5% decrease in antibiotic consumption between 2011 and 2016, and swine exposure levels have continued to decrease in recent years (in 2020, −23.8% from the ALEA value of 2016) [34]. However, antibiotic exposure remains high in pig production, showing an ALEA of 0.46 in 2021 [35]. In our study, we found an ALEA mean of 0.91 in P1 and 0.73 in P2. Two main reasons could explain this difference: a geographical reason (1) or health status regarding PRRSV (2).

(1)All of the consumption levels used for comparison are measured at a national level. In our case, it is a regional study based only on farms located in Brittany, where the prevalence of PRRSV-1 is high. It would be appropriate to include farms spread all over France for comparison. The mg/PCU indicator is also impacted by this point.(2)Secondly, only PRRSV-1 unstable farms were included in our study, which may explain the higher level of antibiotic use due to their poorer health status. Indeed, Trevisi et al. had previously shown that PRRSV status significantly influenced the use of antibiotics in the weaning and fattening stages [36].

Choosing farms according to their level of consumption in P1 would have allowed us to classify them into high, medium, and low consumers, independent of the year of inclusion. A more detailed study of the relationship between the consumption levels and antibiotic usage reduction would then have been available. However, in view of the changing trend of antibiotic use in pig farming, the classification of farms could have differed depending on the year. Combined with the limited number of farms and the different years of inclusion in our study, such a classification was therefore not possible.

Finally, a superior study protocol would include control farms (i.e., unstable with no implementation of control measures) to allow a comparison between farms that carried out a successful PRRSV-1 stabilization program. However, this was not feasible with regard to ethics and welfare.

## 5. Conclusions

After careful interpretation of our results and considering the limitations of the study, we showed that the implementation of a PRRSV-1 stabilization program might help to reduce antibiotic usage, independently of the indicator used (ALEA or mg/PCU). To the best of our knowledge, it is the first study showing the impact of such stabilization programs on antibiotic usage. Together with an economical approach (return on investment of such protocols), the supplementary benefits of reducing antibiotic usages are important data to be considered. These promising results show that further investigations into the benefits of PRRSV-1 control measures and antibiotic usage in swine herds could be beneficial, as such measures can positively impact the reduction in antibiotic use.

## Figures and Tables

**Figure 1 animals-13-02270-f001:**
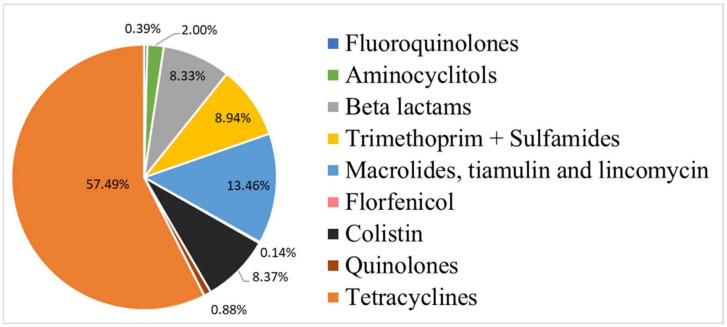
Distribution of antibiotic families regarding the quantities of active ingredients consumed during both periods (19 farms).

**Figure 2 animals-13-02270-f002:**
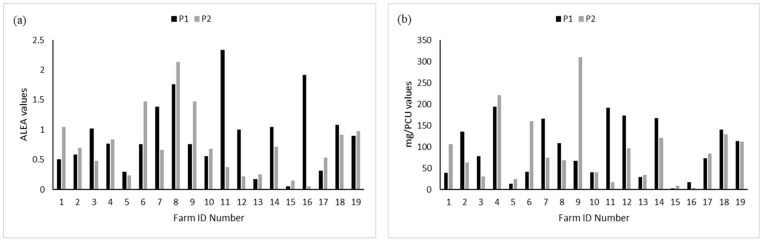
Overall consumption per farm between P1 and P2 for (**a**) ALEA and (**b**) mg/PCU. The bars represent the sum of all treatments implemented during each period.

**Figure 3 animals-13-02270-f003:**
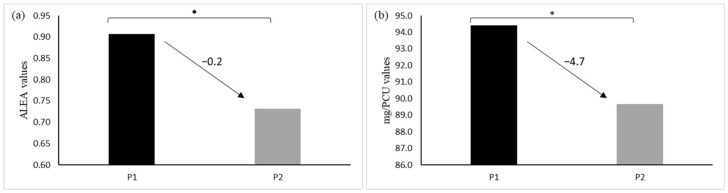
Comparisons of mean antibiotic consumption between P1 and P2 measured as ALEA (**a**) and as mg/PCU (**b**). The bars represent the average value of antibiotic usage obtained from all farms during the two periods. Two datasets for each indicator (ALEA and mg/PCU) were analyzed by Kruskal-Wallis test; ‘*’ means *p* < 0.05 and ‘•’ means *p* < 0.10.

**Figure 4 animals-13-02270-f004:**
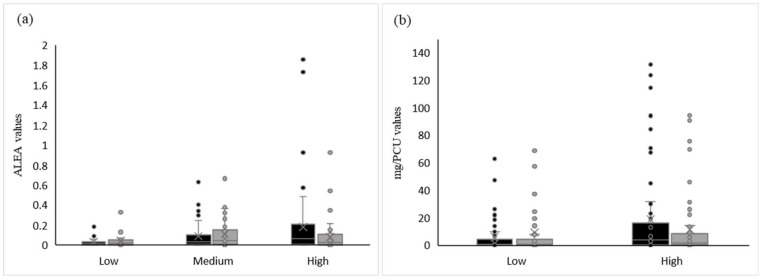
Distribution of antibiotic usage values using two different indicators: ALEA, a French indicator (**a**), and the European indicator expressed in mg/PCU (**b**). Black blocks show antibiotic usage in P1; grey blocks usage in P2. The horizontal line in each block represents the median.

**Table 2 animals-13-02270-t002:** Selected farm characteristics.

Number of Farms	19 Farrow-to-Finish Farms
Average herd size in number of sows (range)	273 (100–600)
Number of farms per batch management:	
Three batches	1
Five batches	1
Seven batches	14
Ten batches	2
Twenty batches	1
Age at weaning:	
21 days	4
28 days	15

**Table 3 animals-13-02270-t003:** Quantity of antibiotics administered to animals over both periods per treatment type.

		Average Antibiotic Usage per Treatment			
		Absolute Quantity		Exposure Indicators	
Numbers of Treatments	Treatment Types	mg	Percent	ALEA ^1^ ± SD ^3^	mg/PCU ^2^ ± SD ^3^
Period 1					
Treatments (*n* = 256)	Individual	234.6	4%	0.03 ± 0.11	0.6 ± 1.2
	Collective	5934.6	96%	0.13 ± 0.19	17.7 ± 27.6
Period 2					
Treatments (*n* = 311)	Individual	167.6	3%	0.02 ± 0.04	0.4 ± 0.8
	Collective	4971.7	97%	0.10 ± 0.12	13.7 ± 26.5

^1^ ALEA (Animal Level of Exposure to Antimicrobials), the French indicator, expresses the ratio between the liveweight receiving a treatment course and the total biomass that could have been treated. ^2^ mg/PCU, defined as the amount of active substance/population correction unit (PCU). ^3^ SD = Standard Deviation.

## Data Availability

The study did not report any data.

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
