# Peer review of "PRRSV-1 Stabilization Programs in French Farrow-to-Finish Farms: A Way to Reduce Antibiotic Usage"

_animals, 2023, doi:10.3390/ani13142270_

Round 1
Reviewer 1 Report
Comments for the authors
The manuscript is nicely done and written. The study design is appropriate and apparently, the analyses were carefully performed. I believe that the results are valuable for the scientific community and has significant scientific merit, as it will probably ignite many further studies in the near future. However, some points need to be clarified before the publication.
Introduction
Well-written introduction.
Materials and Methods
- provide more data on the PRRSV-1 stabilization program implemented by the trial farms (specific day of mass vaccination of sows and piglets)
- The piglets in each pen had possible physical contact with piglets from another pen, both within treatment or from different treatments?
Discussion:
- you could discuss the economic impact of your results on pig production.
Minor comments
Please check the uploaded pdf file.

Minor editing of English language required.
Author Response
Many thanks for your positive comments
- Provide more data on the PRRSV-1 stabilization program implemented by the trial farms (specific day of mass vaccination of sows and piglets.
Farms included in the study should have implemented a successful PRRSV-1 stabilization programme. This consisted of mass vaccination of sows and their piglets (post-weaning and in some cases finishing ones) with a MLV vaccine, together with herd closure and unidirectional pig and human flows. More precisely, it's a combination of a mass vaccination of sows, gilts, and piglets with a PRRSV-1 vaccine (two times at one month interval) followed by batch to batch vaccination of weaning piglets (2 times at 3 weeks interval) during around 6 months, temporary herd closure and strict internal biosecurity measures. All steps of the implementation of PRRSV-1 stabilization programmes are described in Berton et al., 2017 [10]. Based mainly on blood samples from weaning piglets, the success of such protocols was confirmed by the absence of viral circulation in the breeding herds following AASV guidelines [11].
We have precise these elements in the manuscript L 92-96.
- The piglets in each pen had possible physical contact with piglets from another pen, both within treatment or from different treatments?
In this study, only total quantities were recorded (including individual and collective treatments in all physiological stages) to understand if PRRSV stabilization protocols can impact antibiotic usage. We were not able retrospectively to be more precise in the records. But of course, as we discuss at the end, develop other studies to focus on each treatment and their use can be very attractive.
- You could discuss the economic impact of your results on pig production.
Of course, it would be interesting if the economic dimension had been studied but it was not the goal of this research which only focused on antibiotic usages evolution.

Reviewer 2 Report
This manuscript is well-written and thoughtful. Between 2007 and 2019, the authors looked into 19 French farrow-to-finish farms that had effectively adopted the PRRSV-1 stabilizing regimen. For each farm, antibiotic intake (represented in mg/PCU and ALEA) was compared one year before (P1) and one year after (P2) the implementation of such a program, and the difference between P1 and P2 was calculated in percentages. The change from P1 to P2 demonstrated a correlation between higher consumption in P1 and a more significant reduction in antibiotic use in P2. In particular, on farms with higher antibiotic use, the study highlights how stability against PRRSV-1 can reduce antibiotic use.
I have one minor comment:
Figure 1 can include P1 and P2 antibiotic consumption in the 19 farms.
Author Response
Many thanks for your positive comments.
If needed, we have already done this graph, we can add it as supplementary material if you think it beneficial.

Reviewer 3 Report
Overall Comments:
In this manuscript, Costa and co-authors investigate the impact of Porcine Reproductive and Respiratory Syndrome Virus type 1 (PRRSV-1) stabilization programs on reducing antibiotic usage in French farrow-to-finish farms. The study was conducted on 19 farms that implemented a PRRSV-1 stabilization protocol between 2007 and 2019. The data showed that farms with higher antibiotic usage levels before the implementation of the stabilization protocol had a greater reduction in antibiotic usage after its introduction. The study concludes that PRRSV-1 control measures through a stabilization protocol reduced the need for antibiotic usage in the studied farms. However, the study has several limitations, including a small sample size and the absence of control farms. Overall, the manuscript could benefit from extensive revisions before considered for publication.
Major comments:
1) The main weaknesses of the article are the limited number of farms included and the absence of control farms to compare the effect of PRRSV-1 stabilization programs on antibiotic usage reduction.
2) Furthermore, there was no classification of farms according to their level of consumption in P1, which could have provided more information on the relationship between consumption levels and antibiotic usage reduction, and the absence of this classification made the comparison of farms over different years not possible.
3) Finally, the article did not provide a detailed investigation of the relationship between PRRSV-1 control measures and antibiotic usage in swine herds, which could have provided more comprehensive information on the benefits of such measures.
4) Abstract: In this section, authors should stress what methods have been adopted to carry out the research, what results have been obtained, and what conclusions have been drawn in the end. Thus, authors should strengthen this part because it is weak at its current state. I suggest rewrite this section.
5) Reference list: The format of some references is incomplete, and some lack page numbers. Please check carefully throughout the manuscript.
6) This draft is in need of significant language editing due to its poor structure and writing.
Minor comments:
7) Line 166, The title should be 3.1.2.
8) Line 193, The title should be 3.2. All the titles below are incorrect, please correct this issue throughout the manuscript.
9) Line 68, please supplement the ATCC No. of the Vero cell.
Additional questions:
1) How were the antibiotic consumption levels measured and compared in the study?
2) Are there any potential drawbacks or challenges to implementing the protocol in other farms or regions? This issue should address in the manuscript.
This draft is in need of significant language editing due to its poor structure and writing.
Author Response
- The main weaknesses of the article are the limited number of farms included and the absence of control farms to compare the effect of PRRSV-1 stabilization programs on antibiotic usage reduction.
Dear reviewer, for sure, you are right, and we are aware that it is a bias in this study and we discussed it in the manuscript.
With this study, we aimed to present the impact of PRRSV stabilization programmes on these 19 farms as a proof of concept that requires further study to confirm our result in more farms but at this time, we only had these herds that fulfilled inclusion criteria.
Regarding the absence of control, we also agree with you but it is really difficult to have control farms. Indeed, this would mean to let farms with a severe outbreak and circulation of PRRSV without implementing anything to control it.
Despite these observations, we believe that it would be interesting to publish our data for swine veterinarians.
- Furthermore, there was no classification of farms according to their level of consumption in P1, which could have provided more information on the relationship between consumption levels and antibiotic usage reduction, and the absence of this classification made the comparison of farms over different years not possible.
Dear reviewer, could you precise your question because we are not sure to understand it, indeed we made a classification of farms regarding their uses in P1 (Lines 117-122).
- Finally, the article did not provide a detailed investigation of the relationship between PRRSV-1 control measures and antibiotic usage in swine herds, which could have provided more comprehensive information on the benefits of such measures.
Dear reviewer, for sure, this study calls for more investigations which could be conducted aiming to better understand usages and treatments policies in farm. For our study, regarding the fact that it was retrospective, we only could use the global records and we were not able to be more precise.
- Abstract: In this section, authors should stress what methods have been adopted to carry out the research, what results have been obtained, and what conclusions have been drawn in the end. Thus, authors should strengthen this part because it is weak at its current state. I suggest rewrite this section.
Thank you for your comment. We tried to improve our abstract as you asked for (L 19-27).
- Reference list: The format of some references is incomplete, and some lack page numbers. Please check carefully throughout the manuscript.
- We made corrections as asked.
- Line 68, please supplement the ATCC No. of the Vero cell.
Dear reviewer, we are sorry, but we don’t understand what do you mean?
- How were the antibiotic consumption levels measured and compared in the study?
All details concerning the antibiotic consumption measured are included in section 2.2. (L 124-143).
- Are there any potential drawbacks or challenges to implementing the protocol in other farms or regions? This issue should address in the manuscript.
Yes, probably our data together with a more economical approach should be addressed to the swine industry to develop such beneficial protocols. We added a sentence in the conclusion part L 345-347.

Round 2
Reviewer 3 Report
The authors efficiently stressed my concerns. I agree that it can be published in its current state.